# COVID-19 vaccines uptake: Public knowledge, awareness, perception and acceptance among adult Africans

John K. Ahiakpa[1,2¶]*, Nanma T. Cosmas[3¶], Felix E. Anyiam[4¶], Kingsley O. Enalume[5¶], Ibrahim Lawan[6¶], Ijuptil B. Gabriel[7], Chinonyelum L. Oforka[8], Hamze G. Dahir[9], Salisu T. Fausat[10¶], Maureen A. Nwobodo[11¶], Getrude P. Massawe[12¶], Adachukwu S. Obagha[13¶], Debra U. Okeh[14¶], Benjamin Karikari[2,15¶], Samuel T. Aderonke[16¶], Olushola M. Awoyemi[17¶], Idowu A. Aneyo[8¶], Funmilayo V. Doherty[18¶]

1 Research Desk Consulting Limited, Kwabenya-Accra, Ghana, 2 Organisation of African Academic Doctors, OAAD, Nairobi, Kenya, 3 Department of Medical Microbiology, Faculty of Clinical Sciences, College of Health Sciences, University of Jos, Jos, Nigeria, 4 Centre for Health and Development, University of Port Harcourt, River State, Nigeria, 5 Department of Electrical and Electronic Engineering, Federal University of Petroleum Resources, Effurun, Delta State, Nigeria, 6 School of Biology, University of St Andrews, St. Andrews, United Kingdom, 7 Yola Department of Biochemistry School of Life Sciences, Modibbo Adama University, Girei, Adamawa State, Nigeria, 8 Department of Zoology, University of Lagos, Akoka-Yaba, Lagos, Nigeria, 9 School of Public Health and Nutrition, Amoud University Borama, Awdal, Somaliland, 10 Department of Zoology and Environmental Biology, Olabisi Onabanjo University, Ago-Iwoye, Ogun State, Nigeria, 11 Public Health Department of Gregory University Uturu, Abia State, Nigeria, 12 The Open University of Tanzania, Dar es Salaam, Kinondoni, Tanzania, 13 African Regional Postgraduate Programme in Insect Science, University of Ghana, Legon, Ghana, 14 Department of Community Medicine, Federal Medical Centre Umuahia, Abia State, Nigeria, 15 Department of Crop Science, University for Development Studies, Tamale, Ghana, 16 Department of Biochemistry, Faculty of Basic Medical Sciences, University of Lagos, Lagos, Nigeria, 17 Department of Environmental Toxicology, The Institute Environmental and Human Health, Texas Tech University, Lubbock, Texas, United States of America, 18 Yaba College of Technology, Lagos, Nigeria

¶ Membership of the INASP/AuthorAID Online Journal Club is provided in the Acknowledgments.
* jonahiakpa@outlook.com

**Data Availability Statement:** All relevant data are within the paper and its Supporting Information files.

**Funding:** The authors received no specific funding for this work.

## Abstract

### Introduction

The willingness of Africa's population to patronise the COVID-19 vaccines is critical to the efficiency of national immunisation programmes. This study surveys the views of adult African inhabitants toward vaccination and the possibility of participating or not participating in governments' efforts to get citizens vaccinated.

### Method

A cross-sectional online survey of adult Africans was undertaken from December 2020 to March 2021. Responses were anonymised. The Pearson Chi-square test was performed to determine whether or not there were any variations in knowledge, awareness, perception and acceptance of the COVID-19 vaccines among the participants. Binomial logistic regression was used to evaluate the factors associated with willingness to accept the COVID-19 vaccines and participate in immunisation programmes.

**Competing interests:** The authors have declared that no competing interests exist.

## Results

The results indicate that COVID-19 vaccines are more likely to be used by adult Africans over the age of 18 who are largely technologically savvy (55 percent) if the vaccine is made broadly available. A total of 33 percent of those who responded said they were unlikely to receive the vaccine, with another 15 percent stating they were undecided. Aside from that, we found that vaccine hesitancy was closely associated with socio-demographic characteristics such as age, gender, education and source of information. We also found that there were widespread conspiracies and myths about the COVID-19 vaccines.

## Conclusion

More than one-third of African adults who participated in the survey indicated they would not receive the COVID-19 vaccine, with majority of them expressing skepticisms about the vaccine's efficacy. It is possible that many of the people who would not be vaccinated would have an impact on the implementation of a COVID-19 immunisation programme that is meant for all of society. Majority of the respondents were unwilling to pay for the COVID-19 vaccines when made available. An awareness campaign should be focused on promoting the benefits of vaccination at the individual and population levels, as well as on taking pre-emptive actions to debunk misconceptions about the vaccines before they become further widespread.

## Introduction

Infectious diseases have caused untold suffering around the world. Novel pathogenic infections have triggered numerous disease outbreaks and epidemics on the planet in recent decades. SARS-CoV-2, a new strain of coronavirus from Wuhan, China, sparked the world's vilest pandemic ever [1]. Due to its global reach, it was first labeled an epidemic before being upgraded to a pandemic and finally an infodemic [2]. On February 11, 2020, the World Health Organisation (WHO) named it coronavirus disease 2019 (COVID-19) [3]. With 96% genomic identity to the horseshoe bat virus RaTG13, *Rhinolophus affinis*, SARS-CoV-2 is an enclosed, single-stranded positive-sense RNA virus. A 5'UTR, followed by ORF1a and ORF1ab, four structural genes (spike S, envelope E, membrane M, nucleocapsid N) and accessory proteins are all found in the SARS-CoV-2 genome, which has a total length of 30,000 nucleotides [2]. It is through the angiotensin converting enzyme-2 (ACE-2) receptor that the S gene encodes the well-known homotrimeric, type I fusion and transmembrane glycoprotein that virus entrance into the host target cell is enabled [3, 4]. The virus penetrates the host cell only when two membranes are fused together [4, 5]. The cellular type II transmembrane serine proteases (TMPRSS2) are activated by SARS-CoV-2, which uses ACE-2 as an entrance receptor (TMPRSS2) [6–8]. The host cell's priming of spike protein is crucial for viral entrance. The effectiveness of SARS-CoV-2 transmission is determined by this interaction with the ACE-2 [2, 9]. Infection and transmission of ACE2 cells in the upper respiratory tract can be increased by exploiting a cellular attachment enhancing factor identified and anticipated in novel mutations [10].

In the ever-growing list of dangerous new agents, SARS-CoV-2 is the most recent. It is difficult to determine the number of asymptomatic COVID-19 infected persons [4, 5]. The incubation period for COVID-19 infection is estimated at 2–24 days [9, 11, 12], and symptoms such

as fever, cough, headache, muscle aches, and dyspnoea are usually observed in infected individuals. Patients with unusual signs and symptoms, such as vomiting and diarrhoea, have been observed on rare occasions. Global mortality from COVID-19 was reported by WHO as 3.4% [3]. Over 3.54 billion people have received at least one dose of the SARSCoV-2 vaccines regardless of brand name as of July 20, 2021. However, this pandemic has become a race between vaccine efficacy and new variants.

Many countries' healthcare system has been strained, and with job losses across industries as a result of this pandemic, which have had unquantifiable economic repercussions [9, 10]. Different vaccines have been developed, but the number of confirmed cases and deaths are still rising despite these efforts to stop the spread of the disease. The focus has been placed on the necessity to have people vaccinated with WHO-approved COVID-19 vaccines. Previous studies have shown that vaccination is an effective means of preventing infectious diseases [6]. However, acceptance of vaccines by people does not always translate into vaccine efficacy and availability. Vaccine hesitancy has been attributed to increasing vaccine misinformation which has markedly contributed to the continuous decrease in vaccine uptake globally, leading to the third and fourth waves of the COVID-19 pandemic [7, 8, 13].

Some of the COVID-19 vaccines developed were made utilising four unique methodologies which incorporates viral vector, whole virus, ribonucleic acid (RNA) and protein subunits [5]. Despite the fact that the COVID-19 jabs were developed more rapidly than previous vaccines, they have been meticulously tried and tested [11]. Vaccine acceptance amongst the overall population and medical workers have a crucial role in the control of the pandemic. The COVID-19 vaccines are efficacious in preventing COVID-19; however, their effectiveness and viability is dependent upon dosage, seriousness of disease, and COVID-19 variation. For instance, the Pfizer-BioNTech vaccine is estimated at 95%, Moderna 94.1% and Janssen 66.3% efficacies [4, 12, 14].

Alpha variant infections are effectively treated with the mRNA COVID-19 vaccines from Pfizer-BioNTech and Moderna. Sera from a Pfizer-vaccinated health care worker was found to be effective in neutralising B.1.1.7 [9, 15]. The Johnson and Johnson single shot is reported to be quite effective in producing protective neutralising antibodies. Moderna and Novavax vaccines were found to be less effective at neutralising antibodies. Vaccines from Pfizer-BioNTech and Moderna showed no change in neutralisation of S447N, but lowered neutralisation of E484K. Vaccines designed to protect against Beta strains are less effective than those designed to protect against other strains. There was only 75% efficacy for Pfizer's vaccine in clinical trials [16, 17], whereas in the South African trials, the AstraZeneca AZD1222 vaccine failed to prevent even mild or moderate COVID-19 infection [18]. Good neutralisation was seen with the Covaxin and NVX-CoV2373 (Covavax) vaccines. The vaccines from Pfizer-BioNTech and Moderna have a lower level of neutralising antibodies.

In a pre-clinical vivo research, the monoclonal antibody regdanvimab (CT-P59) displayed significant neutralising activity against the Delta variant, B.1.617.2, as well as against the Lambda variant in a cell-based pseudovirus assay. Among antivirals, the most widely used drug is remdesivir which inhibits the viral RNA dependent RNA polymerase and has since been approved by US-FDA for adults and paediatric patients with severe symptoms. China treated 85% of the COVID-19 patients using traditional medicines such as root extract of *Isatis indigotica* and extract of *Houttuynia cordata* [9, 10, 13].

Roughly, 80–89% of vaccinated individuals show low rate of local symptoms and 55–83% shows as a minimum of one systemic symptom following immunisation [19]. However, evaluation of attitudes and acceptance rates towards COVID-19 vaccines can shape communication campaigns that are much needed to reinforce trust in vaccination programmes [8]. Vaccination is perhaps the most sustainable intervention to forestall COVID-19 infections [3]. The

quickest a vaccine had at any point been created before this pandemic was four years [3, 14, 20], but COVID-19 vaccines were developed under one year. Vaccine hesitancy mirrors public health hazard [6, 21]. The Strategic Advisory Group of Experts on Immunisation (SAGE), defines vaccine hesitancy as the "delay in acceptance or refusal of immunisation regardless of accessibility of immunisation service" [6]. Vaccine hesitancy originates from perceived risks versus benefits, certain strict religious convictions and absence of credible information and mindfulness [11, 16, 22, 23], and negative perceptions towards COVID-19 vaccines [21, 24].

In Africa, vaccine hesitancy is premised on perceived danger of the vaccines, safety and effectiveness of vaccines, general immunisation approach, previous immunisation experiences, religious beliefs, immunisation accessibility and socio-cultural constraints [21, 25]. A survey by Lazarus *et al.* [7] revealed vaccine acceptance rate of 81.6% in South Africa and 65.2% in Nigeria. A study on early awareness, perception and practices towards COVID-19 vaccines from North-Central Nigeria showed an acceptance rate of 29.0% [26]. Public health communication needs to assure people of the COVID-19 vaccines safety and their benefits. Awareness of COVID-19 vaccines will play a key role in maintaining the public confidence in vaccination [15, 25]. This will require effective communication through adequate resources and planning. Public announcements, advertisement, jingles, webinar, workshops, and trainings are needed to be in place as early as possible and continue until full vaccination is achieved since COVID-19 vaccines are now available. This will provide transparent information against rumours and conspiracy theories. Prior knowledge of vaccination shows that most people on the average could be willing to accept the COVID-19 vaccines with less side effects. However, awareness campaigns could increase the readiness for COVID-19 vaccination programmes across Africa [21].

A key factor in low vaccine acceptance is exposure to misinformation and conspiracy theories. Hesitancy to the COVID-19 vaccines could impede the success of vaccination programmes [17, 18, 27–29]. Also, the speed of COVID-19 vaccine development, registration and deployment in less than a year have contributed to the level of hesitancy in Africa [7].

## Materials and methods

### Design of the study and participants

Using a random selection process, this online cross-sectional survey was conducted at the continental level with randomly selected participants. The interviews were undertaken between December 2020 and March 2021 with the assistance of collaborators from each of the participating countries. A questionnaire with 33 question items, separated into four sections, was created. After answering a few demographic questions (such as where you live and what you do for a living), respondents were asked a series of questions about their medical and past immunisation history. Our definition for adult Africans refers to Africans aged from 18 years and above. The remaining three sections described COVID-19 vaccine's history and how it's administered. Collaborators and the study team reviewed the survey a number of times. Using a piloted sample of 30 people, we tested the questions' reliability and how long it took to interview one person. To ensure proper data collection and storage, members of the research team reviewed the data several times a day.

The survey questionnaire was created in Microsoft Forms, and was sent by email, and via social media platforms such as Facebook, LinkedIn, Twitter, Telegram, WhatsApp, WeChat and other social media platforms. In this study, participants volunteered their time and were not compensated in any way for their participation. All responses were treated as entirely confidential and were not shared with anyone. In order to reach literate Africans with online presence, we employed virtual networks to reach the general public using the snowballing or

chain-referral approach, which saves us both time and money [21]. Even though the representativeness of our survey is compromised by selection bias, we believe that reaching out to Africa's online population is a worthwhile endeavour because vaccine hesitancy among Africa's "literate" population has significant ramifications for the rest of the continent's population [16, 25]. Social media such as WhatsApp, Facebook and Twitter are popular social media platforms where misinformation, and fake news are communicated and transmitted. Thus, sampling public opinion through these networks, is critical for public health planning [13, 17, 18]. Those who frequent the internet are more likely than others to be linked to networks outside their immediate locations (particularly abroad) and to be affected by online vaccine conspiracies coming from remote locations. Adults without internet access may be persuaded to get the vaccine by those on social media or by word-of-mouth. Our study reporting was done in accordance with the STROBE guidelines [27].

## Statistical analyses

Public health specialists with many years of experience in conducting surveys were consulted in the development of the questionnaire [29]. A test group of 20 people took part in the questionnaire before it was rolled out to the public, but they were not included in the final survey. The conventional Cochran formula [29] was used to determine the starting sample size;

$$no = \frac{Z2pq}{e2},$$

where e = the desired precision level (margin of error), where p is the fraction of population, q is 1-p, and Z is the Z-value found in a Z table. A total of 365 participants completed the closed-ended questionnaire for our study. At a 95% level of confidence, this corresponds to a 2% margin of error [29].

Descriptive statistics were employed to summarise the survey data and describe the socio-demographic characteristics of the study participants. Chi-square tests were then used to estimate the correlations between socio-demographic variables and participants' willingness to receive a COVID-19 vaccine. Variables such as likelihood (very likely or somewhat likely), mix (not decided), or negative (somewhat unlikely or very unlikely) responses to the COVID-19 vaccine were trichotomised to compare responses for various socio-demographic characteristics. A statistically significant p-value of 0.05 and an alpha level of 5 percent were used to assess potential vaccine hesitancy.

## Ethical considerations

Ethical clearance was obtained from the School of Postgraduate Studies and Research, Amoud University in Somalia. Prior to the data collection, participants were required to provide written consent at the time of data collection. Each participant was asked to sign the form to attest that they had voluntarily chosen to participate in the study. It was made clear that anyone who did not wish to engage in the study had the option to do so. Throughout the survey process, participants' responses were kept completely confidential. All dataset was de-identified to ensure no participant's identity was revealed.

## Results

### Socio-demographic characteristics of respondents

An overview of the demographic profiles of the 365 survey respondents is presented in Table 1 below. The age distribution of the respondents ranged from 65 and above (n = 9; 2.47%) to

**Table 1. Demographic information of respondents (n = 365).**

| Variable | Frequency (n) | Percent (%) |
|---|---|---|
| **Age** | | |
| 18–29 | 169 | 46.30 |
| 30–49 | 158 | 43.29 |
| 50–64 | 29 | 7.95 |
| 65 & Above | 9 | 2.47 |
| **Sex** | | |
| Male | 208 | 56.99 |
| Female | 157 | 43.01 |
| **Marital Status** | | |
| Single | 201 | 55.07 |
| Married | 157 | 43.01 |
| Widow(er) | 6 | 1.64 |
| Divorced | 1 | 0.27 |
| **Highest Educational Level attended** | | |
| Basic/Primary school | 1 | 0.27 |
| Secondary/High school | 12 | 3.29 |
| Diploma | 14 | 3.84 |
| Bachelor's Degree | 182 | 49.86 |
| Master's and Above | 156 | 42.74 |
| **Occupation** | | |
| Student | 93 | 25.48 |
| Health care worker | 69 | 18.90 |
| University lecturer/researcher | 64 | 17.53 |
| Civil servant | 38 | 10.41 |
| Business man/woman | 28 | 7.67 |
| Professional (Engineer, Accountant, consultant) | 21 | 5.75 |
| Administrator | 19 | 5.21 |
| Teacher | 13 | 3.56 |
| Others | 33 | 29.37 |
| **Country of Origin** | | |
| Nigeria | 174 | 47.67 |
| Somalia | 111 | 30.41 |
| Ghana | 38 | 10.41 |
| Mozambique | 15 | 4.11 |
| Kenya | 5 | 1.37 |
| Ethiopia | 4 | 1.10 |
| Rwanda | 4 | 1.10 |
| Tanzania | 3 | 0.82 |
| Zambia | 3 | 0.82 |
| Uganda | 2 | 0.55 |
| Malawi | 1 | 0.27 |
| Morocco | 1 | 0.27 |
| Botswana | 1 | 0.27 |
| Congo, Republic of the | 1 | 0.27 |
| Djibouti | 1 | 0.27 |
| Eswatini (formerly Swaziland) | 1 | 0.27 |
| **Country of Residence (n = 352)** | | |

*(Continued)*

**Table 1.** (Continued)

| Variable | Frequency (n) | Percent (%) |
|---|---|---|
| **Age** | | |
| Nigeria | 169 | 48.01 |
| Somalia | 104 | 29.55 |
| Ghana | 37 | 10.51 |
| Mozambique | 15 | 4.26 |
| Kenya | 4 | 1.14 |
| Rwanda | 4 | 1.14 |
| Tanzania | 3 | 0.85 |
| Uganda | 3 | 0.85 |
| Zambia | 3 | 0.85 |
| Ethiopia | 2 | 0.57 |
| South Africa | 2 | 0.57 |
| Malawi | 3 | 0.85 |
| Morocco | 3 | 0.85 |
| Botswana | 3 | 0.85 |
| Cameroon | 2 | 0.57 |
| Democratic Republic of the Congo | 2 | 0.57 |
| Eswatini (formerly Swaziland) | 2 | 0.57 |

18–29 (n = 169; 46.30%); indicating the youthfulness of the respondents (Table 1). In terms of gender, the proportion of male participants in the study was 56.99% compared to 43.01% of female participants in the survey. Majority of the participants were single (55.07%; n = 201); while 43.01% (n = 157) were married. On educational attainment, majority of the participants have a university degree (49.86%; n = 182), while 3.56% (n = 13) had basic or secondary school. We also profiled the occupation of the participants. Majority of the participants were students (n = 95; 25.48%) while 3.56% (n = 13) were teachers (Table 1). Majority of the participants were Nigerians while the country with the least participation were Malawi, Morocco, Botswana, Cameroon, Democratic Republic of the Congo and Eswatini (Table 1).

The socio-demographic characteristics of respondents revealed that social media campaigns yielded the highest awareness (90.4%), local TV/radio (86.9%), newspaper (60%), community mobilisation (4.1%), religious gatherings (3%) and courses/flyers (0.6%) (Fig 1A). However, the respondents considered social media as a more accessible platform to disseminate information for all groups of people.

The result also indicated that one third of the respondents (73%) do not show interest in taking the COVID-19 test and about one third of the respondents (27%) have taken COVID-19 test before (Fig 1B). The result of the COVID-19 vaccine acceptability showed variability in the opinions of Africans. The result indicated that about 59% are willing to receive the COVID-19 vaccine, about 22% respondents were outrightly not in support of the COVID-19 vaccine no matter the directive given by their governments while about 19% were indifferent about the vaccine, although this group of people might later change their perspective to receive the vaccine or never (Fig 1C).

The participants showed low awareness (65%) of the COVID-19 pandemic while only about one third (35%) of the respondents demonstrated some level of awareness (Table 2). Respondents were asked to give their opinion on whose responsibility it should be in creating the awareness with multiple choices provided ranging from the government, media outlets, organisations, and individuals. From the survey, 83% of the respondents believed that the onus

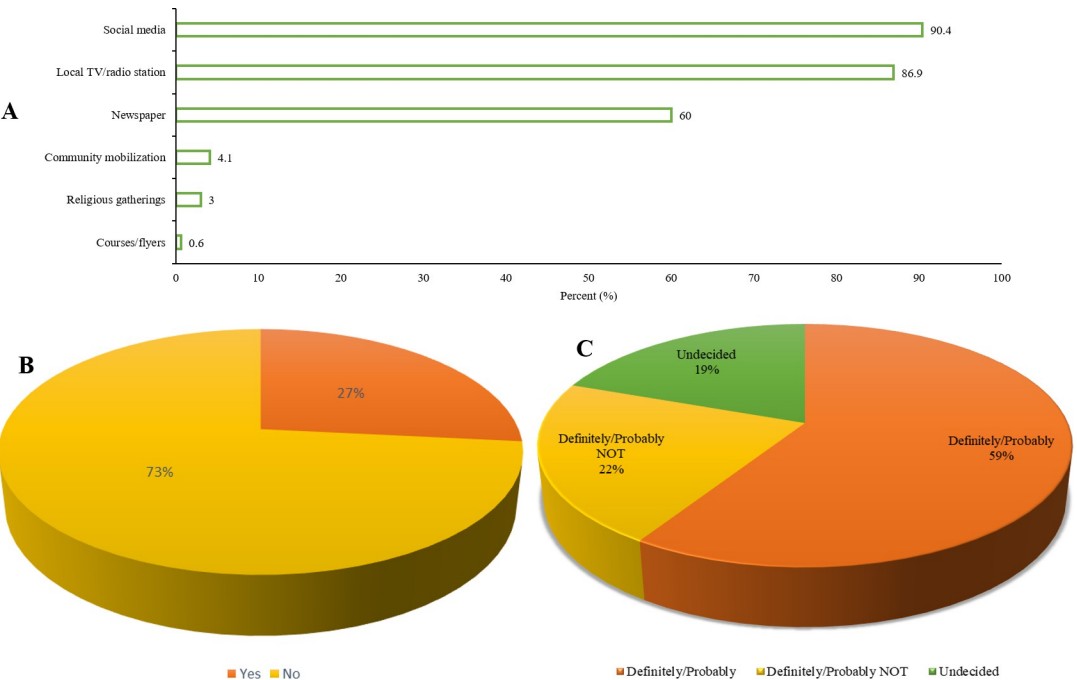

**Fig 1. Effectiveness of media campaigns on COVID-19 vaccine.** (**A**) vaccine awareness campaign with the most reach. (**B**) COVID-19 test by participants. (**C**) level of COVID-19 vaccine hesitancy among adult Africans.

for awareness campaign is on the government, followed by the media (78%), health workers (~76%), WHO (~75%) and about 1% for individuals and community/traditional leaders.

In measuring public knowledge of the COVID-19 vaccines, we asked questions to assess participants' knowledge on the COVID-19 vaccines. About 26.58% (n = 97) of the participants indicated to have been previously diagnosed with COVID-19; while 73.42% (n = 268) of the

**Table 2. Awareness among the general public on COVID-19 pandemic (n = 365).**

| Variable | Frequency (n) | Percent (%) |
|---|---|---|
| Do you think enough awareness has been created about the COVID-19? Γ | | |
| Yes | 126 | 34.52 |
| No | 239 | 65.48 |
| In your opinion who should be involved in the awareness campaign Γ | | |
| Government Γ | 303 | 83.01 |
| Media Γ | 285 | 78.02 |
| Health Workers Γ | 277 | 75.89 |
| World Health Organisation | 272 | 74.52 |
| Religious leaders | 253 | 69.32 |
| Educational/Research Institution | 252 | 69.04 |
| Centers for Disease Control (CDC) | 237 | 64.93 |
| Civil Society Organisations (CSO) | 222 | 60.82 |
| Industry | 151 | 41.37 |
| Individuals | 4 | 1.10 |
| Community/Traditional leaders | 3 | 0.82 |

Γ = Multiple response applies

participants indicated they have not been diagnosed with COVID-19 before. Participants were also asked what action they will likely take when diagnosed with COVID-19; majority (n = 144; 39.45%) revealed they will resort to medications (drugs), 31.23% (n = 114) indicated they will resort to herbal remedies, 21.64% (n = 79) will opt for COVID-19 vaccination (vaccines); while the rest indicated isolation/quarantine, resorting to immune boosting diets, seeking medical attention and prayer as first line of actions (Table 3). Majority of the participants

**Table 3. Public knowledge on COVID-19 vaccinations (n = 365).**

| Variable | Frequency (n) | Percent (%) |
|---|---|---|
| **Have you been diagnosed with COVID-19 before?** | | |
| Yes | 97 | 26.58 |
| No | 268 | 73.42 |
| **If you were diagnosed with COVID-19 what will be your first option?** | | |
| Drugs | 144 | 39.45 |
| Herbal remedies | 114 | 31.23 |
| Vaccine | 79 | 21.64 |
| Isolation/Quarantine | 10 | 2.74 |
| Immune boosting diet | 6 | 1.64 |
| Prayer | 5 | 1.37 |
| Seek medical attention | 2 | 0.55 |
| I don't know | 5 | 1.37 |
| **Would you still get the COVID-19 vaccine if you recovered from COVID?** | | |
| Yes | 197 | 53.97 |
| No | 72 | 19.73 |
| I don't know | 96 | 26.30 |
| **In what order should the COVID-19 vaccination be rolled out?** | | |
| Correct order presented[μ] | 57 | 15.62 |
| Wrong order presented | 308 | 84.38 |
| **Do you agree with hand washing as a COVID-19 Prevention behaviour?** | | |
| Strongly agree/agree | 76 | 20.82 |
| Disagree/strongly disagree | 289 | 79.18 |
| **Do you agree with wearing of nose or face shield as a COVID-19 Prevention behaviour?** | | |
| Strongly agree/agree | 149 | 40.82 |
| Disagree/ strongly disagree | 216 | 59.18 |
| **Do you agree with social distancing as a COVID-19 Prevention behaviour?** | | |
| Strongly agree/agree | 147 | 40.27 |
| Disagree/ strongly disagree | 218 | 59.73 |
| If you have not been vaccinated what can you do to stay safe[γ?] | | |
| Regular hand washing or use of alcohol-based hand sanitiser | 316 | 86.58 |
| Wearing of nose mask or face shield | 297 | 81.37 |
| Maintaining social distancing | 275 | 75.34 |
| Use of local herbal mixtures | 84 | 23.01 |
| Praying | 2 | 0.55 |
| Maintaining healthy diet and lifestyle | 2 | 0.55 |
| Socially observant for people with symptoms of COVID | 1 | 0.27 |

γ = Multiple response applies; μ = Front-line health workers>Individuals age 50 & above > Individuals age 18 to 49 with relevant medical conditions>Government officials & strategic leaders>Individuals age 18 to 49 without relevant medical conditions

(n = 197; 53.97%) indicated they will still accept COVID-19 vaccines even after recovery from an earlier COVID-19 treatment (Table 3). Majority of the participants (n = 308; 84.38%) wrongly ranked the order in which the COVID-19 vaccination should be rolled out at country levels with only 15.62% (n = 57) ranking the vaccine roll out order accurately. Participants disagree/strongly disagree that all COVID-19 prevention protocols such as hand-washing, wearing of face-mask/face shield, social distancing and use of hand sanitisers should continue even after vaccination (Table 3).

## Public perception on the COVID-19 vaccines

The perceptions of the participants across diverse countries in Africa was assessed. Among the 365 participants, 96.44% mentioned that COVID-19 vaccines had arrived in their respective countries as at the time of the study. On the contrary, 3.57% indicated they had no or are not aware of the arrival of the vaccines. Among those who indicated to have knowledge on the arrival of vaccines in their countries (n = 352), 76.14% of them mentioned AstraZeneca vaccine; while 0.28–1.99 stated either Pfizer-BioNTech, Sinopharm, Johnson and Johnson, Sputnik V or Moderna vaccines. The remaining 19.03% participants strikingly indicated not to have idea on the brand of vaccine in their countries (Table 4). This trend suggests that AstraZeneca vaccine is the well-known vaccine in the participating countries in this study.

Interestingly, 44.89% of the 352 participants were of the opinion that vaccine brands purchased/arrived in their respective countries are effective against the deadly virus, 42.05% had no idea on the effectiveness of the vaccines in their countries and the remaining 13.07% stated emphatically that vaccines in their countries are not effective against the virus (Table 4). Among the reasons ascribed to no effectiveness of vaccines in their countries include not certain on its effectiveness (39.13%), the associated side effects (28.26%), and doubts (15.22%). Prior to COVID-19 vaccination, 87.95% of the 365 participants willingly and usually accepted vaccination, but the 12.05% participants would not willingly accept the COVID-19 vaccination due to some personal reasons. Among the reasons for objection to vaccination include doubts (47.73%), side effects (34.09%), healthy condition (2.27%) and do not want to be used as experimental animals (guinea pigs, 2.27%) and remaining 13.64% had no reason for objecting to the vaccination.

Majority of the participants (48.22%, n = 365) were of the opinion that COVID-19 vaccines are safe, while 17.81% participants believed that the vaccines are not safe and 33.97% participants had no knowledge on the safety of the vaccines. With regards to the effectiveness of the vaccines, 44.38, 14.52 and 41.10% indicated that vaccines are effective, ineffective and no knowledge, respectively (Table 4). In addition, 33.97% participants (n = 365) mentioned that COVID-19 vaccines have serious side effects, 32.88% participants opined that the vaccines have no serious side effects and 33.15% participants did not know if the vaccines have any serious side effects. Empirically, 41.37% of the participants (n = 365) had positive perception on the COVID-19 vaccines, while 58.63% participants had negative perception on the vaccines (Table 4) probably due to inadequate public education and several conspiracy theories on the vaccines. These results warrant intensification of public education to counter the numerous conspiracy theories in the public domain.

## Public readiness and willingness to accept COVID-19 vaccines

In measuring overall public readiness for the COVID-19 vaccines, 9 questions were used to assess participants' willingness and readiness. This section was scored 1 for each positive response by the participant, and 0 for each negative response. All answers were summed (as shown in Table 5). Participants with an overall willingness to accept the COVID-19 vaccine

**Table 4. Public perception on the COVID-19 vaccines (n = 365).**

| Variable | Frequency (n) | Percent (%) |
|---|---|---|
| **Has COVID-19 arrived in your country?** | | |
| Yes | 352 | 96.44 |
| No | 4 | 1.10 |
| I don't know | 9 | 2.47 |
| **What type of COVID-19 vaccine is available in your country? (n = 352)** | | |
| AstraZeneca | 268 | 76.14 |
| Pfizer BioNTech | 7 | 1.99 |
| Sinopharm | 4 | 1.14 |
| Johnson & Johnson | 3 | 0.85 |
| Sputnik V | 2 | 0.57 |
| Moderna | 1 | 0.28 |
| I don't know | 67 | 19.03 |
| **Do you think the brand acquired by your country is effective? (n = 352)** | | |
| Yes | 158 | 44.89 |
| No | 46 | 13.07 |
| I don't know | 148 | 42.05 |
| **Reasons you think the brand is not effective? (n = 46)** | | |
| Not certain on its effectiveness | 18 | 39.13 |
| The side effects associated with it | 13 | 28.26 |
| A lot of doubts | 7 | 15.22 |
| Because there is no cure for the COVID-19 | 2 | 4.35 |
| Still an experimental drug | 2 | 4.35 |
| Vaccine was developed so quickly | 2 | 4.35 |
| Don't know which variant of virus it is for | 1 | 2.17 |
| Some countries rejected it | 1 | 2.17 |
| **Do you normally accept vaccination before?** | | |
| Yes | 321 | 87.95 |
| No | 44 | 12.05 |
| **What are some reasons for not accepting vaccination? (n = 44)** | | |
| I have doubts | 21 | 47.73 |
| Side effects | 15 | 34.09 |
| No reason | 6 | 13.64 |
| I am fine and healthy | 1 | 2.27 |
| We are guinea pigs | 1 | 2.27 |
| **Do you think the COVID-19 vaccine is safe?** | | |
| Yes | 176 | 48.22 |
| No | 65 | 17.81 |
| I don't know | 124 | 33.97 |
| **Do you think the COVID-19 vaccine is efficacious (effective)?** | | |
| Yes | 162 | 44.38 |
| No | 53 | 14.52 |
| I don't know | 150 | 41.10 |
| **Do you think the COVID-19 vaccine have serious side effects?** | | |
| Yes | 124 | 33.97 |
| No | 120 | 32.88 |
| I don't know | 121 | 33.15 |
| **Overall public perception of the COVID-19 vaccines** | | |

(*Continued*)

**Table 4.** (Continued)

| Variable | Frequency (n) | Percent (%) |
|---|---|---|
| Positive perception (5–9) | 151 | 41.37 |
| Negative Perception (≤4) | 214 | 58.63 |

were scored 5–9, while those not willing to accept the vaccines were scored ≤4. Participants (58.63%; n = 214) not willing to accept the vaccines were proportionally higher, compared to those willing and ready (41.37%; n = 151^) to accept the vaccines (Table 5). On the other hands, participants were quizzed about the willingness/readiness to pay for the vaccines when available in their countries. Majority of the participants (50%; n = 109) indicated their unwillingness to pay for the vaccines citing several reasons for their unwillingness. Only 49.30% (n = 106) of the participants indicated their willingness to pay for the vaccines. Most of the participants indicated that their governments are naturally expected to provide the vaccines for free (58%; n = 56.31), inability to afford the vaccines (36.89%; n = 38) and skepticism of the efficacy of the vaccines (6.80%; n = 7) to justify their unwillingness to pay for the COVID vaccines (Table 5).

We also profiled common myths and conspiracy theories against the COVID vaccines from the respondents. Interestingly, some of the respondents (41.92%; n = 153) do not subscribe to a conspiracy theory that says the COVID-19 vaccines alter the DNA of recipients; while majority of the respondents (43.29%; n = 158) are uncertain of the veracity of this myth. Again, 14.79% (n = 54) of the respondents however subscribed to this conspiracy theory (Table 6). About 10.96 (n = 10.96) of the respondents think the COVID-19 vaccines contain a tracking device, another weird conspiracy theory making waves on social media. However, majority of the respondents (46.85%; n = 171) disagree with this conspiracy theory; with 42.19% (n = 154) of the respondents uncertain about the validity of this myth. The '*COVID-19 vaccine for Africa is different from that in other continents*' is among the several myths being circulated in several media outlets. Majority of the respondents (36.44%; n = 133) identify this as a conspiracy theory; while 32.60% (n = 119) agree to this as a truth. Respondents generally revealed that one can still contract COVID-19 even after vaccination (46.03; n = 168); while 14.52% (n = 53) indicates it is impossible for a vaccinated person to contract COVID-19 (Table 6).

## Determinants of willingness to accept COVID-19 vaccines among the respondents

Willingness to accept COVID-19 vaccines did not vary across socio-demographic variables, except occupational level that showed a significantly higher willingness to accept COVID-19 vaccines among the retired (p = 0.042) (Table 7).

## General factors associated with the willingness to accept COVID-19 vaccines

Respondents' willingness to accept COVID-19 vaccines varied across selected factors (Table 8). Significantly higher acceptability of COVID-19 vaccine was observed among those that have done COVID-19 test before (p = 0.029), normally would accept vaccine before (p = 0.001), and have a positive perception on the safety and effectiveness of the COVID-19 vaccines (p = 0.001). Significantly lower acceptability was observed among those with a perceived myth on the COVID-19 vaccine containing a tracking device, which could alter DNA and not the same vaccine as the one imported to Africa (p = 0.001).

**Table 5. Public willingness and readiness to accept COVID-19 vaccines (n = 365).**

| Variable | Frequency (n) | Percent (%) |
|---|---|---|
| For what reasons are you accepting the vaccine[γ?] | | |
| It will help me protect my family, friends and others in the community | 138 | 37.81 |
| It will help stop the pandemic | 125 | 34.25 |
| It will prevent me from contracting COVID | 109 | 29.86 |
| The vaccine is safe and effective | 102 | 27.95 |
| It is a requirement for travelling abroad | 64 | 17.53 |
| **Are you willing to pay for the vaccine? (n = 215)** | | |
| Yes | 106 | 49.30 |
| No | 109 | 50.70 |
| **What is your reason for not willing to pay? (n = 103)** | | |
| Government is expected to provide the vaccine for free | 58 | 56.31 |
| I cannot afford to pay for it | 38 | 36.89 |
| I doubt its effectiveness | 7 | 6.80 |
| Why are you rejecting the vaccine? (n = 79)[γ] | | |
| I am not sure the vaccine is clinically safe | 55 | 69.62 |
| I am not sure the vaccine is effective in preventing me from contracting COVID | 43 | 54.43 |
| I am not fully informed about possible side effects of the COVID-19 vaccine | 36 | 45.57 |
| I feel the vaccine in Africa is not the same as the one in other continents, so I don't trust it | 31 | 39.24 |
| I think the vaccine would alter my DNA | 19 | 24.05 |
| I feel it can result in death, especially among the elderly | 18 | 22.78 |
| I feel there is a tracking device in the vaccine | 10 | 12.65 |
| The vaccine is still under investigations | 2 | 2.53 |
| **If more awareness were created and you are satisfied with the safety and efficacy (effectiveness) of the COVID-19 vaccine, would you accept it? (n = 79)** | | |
| Yes | 29 | 36.71 |
| No | 23 | 29.11 |
| Maybe | 27 | 34.18 |
| **If taking the vaccine becomes a necessary requirement for travel, what will you do?** | | |
| Avoid travelling | 45 | 56.96 |
| Take the vaccine | 30 | 37.97 |
| I would protest/sue the imposters of such policies | 2 | 2.53 |
| Undecided | 2 | 1.27 |
| Why are you undecided in accepting the COVID-19 vaccine? (n = 71)[γ] | | |
| I am not sure the vaccine is clinically safe | 43 | 60.56 |
| I am not fully informed about possible side effects of the COVID-19 vaccine | 28 | 39.44 |
| I am not sure if the vaccine in Africa is the same as that in other continents | 18 | 25.35 |
| I am not sure the vaccine is effective in preventing me from contracting COVID-19 | 18 | 25.35 |
| I am not sure if the vaccine would alter my DNA | 13 | 18.31 |
| I feel it can result in death, especially among the elderly | 5 | 7.04 |
| I am not sure if there is a tracking device in the vaccine | 4 | 5.63 |
| I am free from the infection | 1 | 1.41 |

[γ] = Multiple response applies

**Table 6. Myths and socio-cultural perceptions against the vaccines (n = 365).**

| Variable | Frequency (n) | Percent (%) |
|---|---|---|
| Do you think the COVID-19 vaccine will alter your DNA? | | |
| Yes | 54 | 14.79 |
| No | 153 | 41.92 |
| I don't know | 158 | 43.29 |
| Do you think the COVID-19 vaccine contains a tracking device? | | |
| Yes | 40 | 10.96 |
| No | 171 | 46.85 |
| I don't know | 154 | 42.19 |
| Do you think the COVID-19 vaccine for Africa is different from that in other continents? | | |
| Yes | 119 | 32.60 |
| No | 133 | 36.44 |
| I don't know | 113 | 30.96 |
| Do you think one can still get COVID-19 after vaccination? | | |
| Yes | 168 | 46.03 |
| No | 53 | 14.52 |
| Maybe | 144 | 39.45 |

## Modelling factors associated with COVID-19 vaccine hesitancy

As shown in Table 9, significant explanatory variables in the Chi-Square test of association (Table 9) were included for logistic regression analysis. Model I: Non-adjusted (crude) odds ratio (ORs) comprising selected explanatory variable associated with acceptability of COVID-19 vaccine. The study shows a higher OR for the willingness to accept COVID-19 vaccines among those that have done COVID-19 test before (cOR = 2.02, 95% CI; 1.07–3.79, p = 0.029), normally would accept vaccine before (cOR = 5.48, 95% CI; 2.68–11.19, p = 0.001), and have a positive perception on the safety and effectiveness of the COVID-19 vaccines (cOR = 12.81, 95% CI; 5.87–27.94, p = 0.001). A significant lower OR for acceptability was observed among those with a perceived myth on COVID-19 vaccine containing a tracking device (cOR = 0.078, 95% CI; 0.032–0.19, p = 0.001), could alter DNA (cOR = 0.095, 95% CI; 0.043–0.214, p = 0.001) and not the same vaccine as the one brought to Africa (cOR = 0.099, 95% CI; 0.046–0.213, p = 0.001).

Model II: Adjusted ORs comprised selected explanatory variable associated with acceptability of COVID-19 vaccine while controlling for socio-demographic characteristics. After adjusting for confounding variables, only those that have done the COVID-19 test before showed significant higher OR for the willingness to accept the COVID-19 vaccines (aOR = 17.69, 95% CI; 1.21–256.95, p = 0.035). The other variables showed no significant association (p > 0.05).

## Discussion

Vaccine hesitancy can be a significant contributor to the failure to effectively control a pandemic such as the current COVID-19 pandemic [15, 28–30]. Consequently, estimates of vaccine acceptance rates can be useful in planning requisite actions and interventions to raise awareness and reassure people about the safety and benefits of vaccines, which in turn will aid in controlling the spread of the virus and alleviate the negative effects of the pandemic [13, 30–37]. The assessment of attitudes and acceptance rates of COVID-19 vaccines can aid in the development of communication campaigns that are desperately needed to increase public confidence in vaccination programmes [8, 19, 30–32].

**Table 7. Determinants of willingness to accept COVID-19 vaccines (n = 365).**

| Variables | Willingness to accept COVID-19 vaccines Freq (%) | | Total | Chi-square, P-value |
|---|---|---|---|---|
| | Yes N = 215 | No N = 79 | | |
| **Age (years)** | | | | p = 0.614$^\mu$ |
| 18–29 | 100 (70.92) | 41 (29.08) | 141 (100.0) | |
| 30–49 | 88 (72.73) | 33 (27.27) | 121 (100.0) | |
| 50–64 | 19 (82.61) | 4 (17.39) | 23 (100.0) | |
| 65 & Above | 8 (88.89) | 1 (11.11) | 9 (100.0) | |
| **Gender** | | | | $\chi 2 = 0.25$, p = 0.615 |
| Male | 119 (71.69) | 47 (28.31) | 166 (100.0) | |
| Female | 96 (75.0) | 32 (25.0) | 128 (100.0) | |
| **Marital Status** | | | | $\chi 2 = 0.84$, p = 0.359 |
| Single | 127 (70.95) | 52 (29.05) | 179 (100.0) | |
| Married | 88 (76.52) | 27 (23.48) | 115 (100.0) | |
| **Highest Educational Level** | | | | P = 0.117$^\mu$ |
| Primary/Secondary/High school | 9 (90.0) | 1 (10.0) | 10 (100.0) | |
| Diploma | 6 (46.15) | 7 (53.85) | 13 (100.0) | |
| Bachelor's Degree | 110 (72.85) | 41 (27.15) | 151 (100.0) | |
| Master's and Above | 90 (75.0) | 30 (25.0) | 120 (100.0) | |
| **Occupation** | | | | **P = 0.042*$^\mu$** |
| Business man/woman | 19 (90.48) | 2 (9.52) | 21 (100.0) | |
| Civil servant/Administrator | 36 (80.0) | 9 (20.0) | 45 (100.0) | |
| Company worker | 2 (66.67) | 1 (33.33) | 3 (100.0) | |
| Health care worker | 46 (79.31) | 12 (20.69) | 58 (100.0) | |
| Professional | 11 (68.75) | 5 (31.25) | 16 (100.0) | |
| Retired | 5 (100.0) | 0 (0.0) | 5 (100.0) | |
| Student/unemployed | 52 (60.47) | 34 (39.53) | 86 (100.0) | |
| Teacher/University Lecturer/researcher | 44 (73.33) | 16 (26.67) | 60 (100.0) | |

*Statistically significant ($p < 0.05$), $\mu$ = Fishers exact p

We conducted a survey to assess public awareness, vaccine reluctance, and acceptability of COVID-19 vaccines in Africa, as well as the likelihood of participation or non-participation in national government activities to vaccinate persons in each country. According to our data, almost 6 in 10 (55 percent) of mostly urban and adult Africans over the age of 18 years are likely to receive the COVID-19 vaccine if it is made widely available, which is consistent with recent findings by Acheampong et al. [29] in Ghana. As the current study demonstrates, social media is extremely important in promoting public awareness of health-related concerns. We established that it was highly effective using social media to disseminate information about COVID-19 vaccines in Africa. Nonetheless, local television, radio stations, and newspapers have proven to have a larger reach in spreading information about immunisation programmes in many countries, proving to be particularly effective. The findings of Smith et al. [33] who found that social media is an essential tool used by health authorities and governments in promoting public awareness are consistent with this finding. The relevance of various media in keeping the society informed and watchful in respect to public awareness, knowledge, and readiness to participate in the COVID-19 immunisation campaign was also demonstrated in a report by Anwar et al. [34].

**Table 8. General factors associated with the willingness to accept COVID-19 vaccines (n = 365).**

| Variables | Willingness to accept COVID-19 vaccines Freq (%) | | Total | Chi-square, P-value |
|---|---|---|---|---|
| | Yes N = 215 | No N = 79 | | |
| **Have you done COVID-19 test before?** | | | | $\chi 2 = 4.24$, **p = 0.039**[*] |
| Yes | 69 (82.14) | 15 (17.86) | 84 (100.0) | |
| No | 146 (69.52) | 64 (30.48) | 210 (100.0) | |
| **Have you accepted vaccination before?** | | | | $\chi 2 = 23.23$, **p = 0.001**[*] |
| Yes | 200 (78.13) | 56 (21.88) | 256 (100.0) | |
| No | 15 (39.47) | 23 (60.53) | 38 (100.0) | |
| **Positive perception of the safety and effectiveness of the COVID-19 vaccines** | | | | $\chi 2 = 53.77$, **p = 0.001**[*] |
| Yes | 127 (94.07) | 8 (5.93) | 135 (100.0) | |
| No | 88 (55.35) | 71 (44.65) | 159 (100.0) | |
| **The perceived myth on COVID-19 vaccine containing a tracking device** | | | | $\chi 2 = 37.39$, **p = 0.001**[*] |
| Yes | 12 (38.71) | 19 (61.29) | 31 (100.0) | |
| No | 129 (88.97) | 16 (11.03) | 145 (100.0) | |
| **The perceived myths on COVID-19 vaccine altering human DNA** | | | | $\chi 2 = 36.42$, **p = 0.001**[*] |
| Yes | 17 (40.48) | 25 (59.52) | 42 (100.0) | |
| No | 114 (87.69) | 16 (12.31) | 130 (100.0) | |
| **The perceived myths that COVID-19 vaccine for Africa is different** | | | | $\chi 2 = 40.67$, **p = 0.001**[*] |
| Yes | 48 (51.06) | 46 (48.94) | 94 (100.0) | |
| No | 105 (91.30) | 10 (8.70) | 115 (100.0) | |

[*]Significant (p < 0.05)

**Table 9. Factors associated with the willingness to accept COVID-19 vaccines (n = 365).**

| Variables | Model I | | Model II | |
|---|---|---|---|---|
| | cOR [95% CI] | P-value | aOR [95% CI] | P-value |
| **Done COVID-19 test before** | | | | |
| No[R] | | | | |
| Yes | 2.02 [1.07–3.79] | **0.029**[*] | 17.69 [1.21-256-91] | **0.035**[*] |
| **Normally will accept vaccination before** | | | | |
| No[R] | | | | |
| Yes | 5.48 [2.68–11.19] | **0.001**[*] | 4.11 [0.39–43.79] | 0.242 |
| **Positive perception of the safety and effectiveness of the COVID-19 vaccines** | | | | |
| No[R] | | | | |
| Yes | 12.81 [5.87–27.94] | **0.001**[*] | 3.17 [0.33–30.55] | 0.318 |
| **The perceived myth on COVID-19 vaccine containing a tracking device** | | | | |
| No[R] | | | | |
| Yes | 0.078 [0.032–0.19] | **0.001**[*] | 0.10 [0.009–1.07] | 0.057 |
| **The perceived myth on COVID-19 vaccine altering human DNA** | | | | |
| No[R] | | | | |
| Yes | 0.095 [0.043–0.214] | **0.001**[*] | 0.29 [0.031–2.82] | 0.290 |
| **The perceived myth that COVID-19 vaccine for Africa is different** | | | | |
| No[R] | | | | |
| Yes | 0.099 [0.046–0.213] | **0.001**[*] | 0.45 [0.051–3.89] | 0.466 |

[*]Significant (p < 0.05); Notes: R = reference, cOR = crude Odds Ratio, aOR = Adjusted Odds Ratio, Model II: Controlling for Age, Gender, Marital status, educational level and occupation

A further finding of the survey was that about 30% of the participants were unlikely to obtain the vaccine, with another 15% remaining undecided. Variations in vaccine hesitancy, as well as disparities in critical socio-demographic characteristics were also observed. Again, we found that vaccination resistance is low among older age groups, while males are more likely than females to be indecisive about receiving the vaccine. There were no significant relationships found between willingness to receive the vaccines and either education or geographic location in this study. Thus, key stakeholders in the health sector must intensify their efforts in targeted public education and promote knowledge about the individual and societal benefits of vaccinations, particularly among younger populations and with a particular emphasis among men to combat the spread of conspiracies and myths [35–39]. A vaccine information campaign should be aimed at decreasing the dissemination of misleading information about the vaccines.

The general public's attitude toward COVID-19 testing was negative, which could be attributed to the inefficiency of the testing regimes in Africa. Some people believe that the COVID-19 pandemic is not genuine and that it is merely a geopolitical propaganda, despite the fact that there have been several awareness campaigns at all levels. Consequently, they believe that taking the COVID-19 test will not benefit them and that receiving the vaccine will result in health consequences for themselves. Various conspiracies, such as the vaccinations being created for advanced nations, are used to justify their reluctance to receive the vaccines [36, 40]. In many cases, these conspiratorial beliefs have inflamed public suspicion and raised questions about the efficacy of the vaccines, hampering large-scale immunisation campaigns across the continent. In a study of populations in north-central Nigeria, Lazarus et al. [7] found that just 29 percent of those surveyed expressed interest in the COVID-19 vaccines. In addition, a recent Africa CDC [37] report emphasised the importance of addressing issues of faith in vaccines to increase confidence among the public in the management of the COVID-19 pandemic, which is currently ongoing.

Increased public awareness of the COVID-19 pandemic is critical to combating the pandemic and preventing the spread of the deadly viral infection that has claimed millions of lives around the world. It is the responsibility of the appropriate authorities (African governments, Africa CDC, WHO) to effectively and efficiently disseminate appropriate information to the general public in a timely and space-efficient manner [41, 42], as well as with closer collaborations between and among local, state and international agencies to increase public awareness [43–47]. In this way, the risks of infection, health consequences, and identification of the most vulnerable population and/or those suffering from comorbidities might all be communicated in one voice [13, 34, 43, 45–47]. This would also help to minimise the spread of disinformation, misinformation, and conspiracies, as well as facilitate early detection and intervention in the fight against the virus (e.g., vaccination).

It is possible that several participants were excluded from the study because of the lack of stable internet connection, even though data indicates high internet penetration rates and mobile phone use across Africa. This was a cross-sectional study, and thus no causal links can be established between the independent and dependent variables. Additional time points should be included in future survey to further understand how people's attitudes toward vaccination change over time. Policymakers may assess how vaccination hesitancy might change as a result of the emerging mutations of COVID-19.

## Conclusion

COVID-19 vaccination was a 'no-go-area' for less than two-thirds of African adults surveyed, with a proportion of those surveyed expressing doubts on the efficacy of the vaccines. Many of

the people who would not get vaccinated could have an impact on the implementation of a COVID-19 immunisation programme intended for everyone. In order to prevent the harmful effects of their views on others, health ministries should intensify awareness to counter such extreme views against the vaccines. There is a risk that the results of a survey can be interpreted incorrectly because of the method used to distribute questionnaires. Our social media outreach may have excluded many low-income and elderly persons, as well as those with no or minimal education. Consequently, the results of this survey may not be indicative of the desires and hesitancy of the entire African countries that were surveyed. The vaccines were not available at the time of the survey; therefore, the results may have been different from respondents who received a vaccine.

## Supporting information

**S1 Dataset. Raw dataset from survey.**
(XLSX)

## Acknowledgments

We thank all collaborating institutions and partners who circulated the data collection instruments in their networks. We particularly thank the INASP/AuthorAID Journal Club initiative for supporting early career researchers in the global south.

## Author Contributions

**Conceptualization:** John K. Ahiakpa, Nanma T. Cosmas, Hamze G. Dahir, Olushola M. Awoyemi, Funmilayo V. Doherty.

**Data curation:** John K. Ahiakpa, Nanma T. Cosmas, Felix E. Anyiam, Ijuptil B. Gabriel, Debra U. Okeh, Samuel T. Aderonke, Idowu A. Aneyo.

**Formal analysis:** Nanma T. Cosmas, Felix E. Anyiam, Salisu T. Fausat, Maureen A. Nwobodo, Idowu A. Aneyo.

**Investigation:** John K. Ahiakpa, Hamze G. Dahir, Salisu T. Fausat, Maureen A. Nwobodo, Getrude P. Massawe, Adachukwu S. Obagha, Debra U. Okeh, Benjamin Karikari, Samuel T. Aderonke.

**Methodology:** John K. Ahiakpa, Felix E. Anyiam, Kingsley O. Enalume, Ibrahim Lawan, Ijuptil B. Gabriel, Salisu T. Fausat, Maureen A. Nwobodo, Getrude P. Massawe, Adachukwu S. Obagha.

**Project administration:** John K. Ahiakpa.

**Resources:** Chinonyelum L. Oforka, Funmilayo V. Doherty.

**Software:** John K. Ahiakpa, Felix E. Anyiam, Ibrahim Lawan, Ijuptil B. Gabriel, Chinonyelum L. Oforka.

**Supervision:** Funmilayo V. Doherty.

**Validation:** Ibrahim Lawan, Adachukwu S. Obagha, Benjamin Karikari, Olushola M. Awoyemi, Funmilayo V. Doherty.

**Visualization:** John K. Ahiakpa, Nanma T. Cosmas, Felix E. Anyiam, Chinonyelum L. Oforka, Samuel T. Aderonke, Idowu A. Aneyo.

**Writing – original draft:** John K. Ahiakpa, Nanma T. Cosmas, Felix E. Anyiam, Kingsley O. Enalume, Ibrahim Lawan, Hamze G. Dahir, Salisu T. Fausat, Maureen A. Nwobodo, Getrude P. Massawe, Adachukwu S. Obagha, Debra U. Okeh, Benjamin Karikari, Samuel T. Aderonke, Olushola M. Awoyemi, Idowu A. Aneyo.

**Writing – review & editing:** John K. Ahiakpa, Nanma T. Cosmas, Felix E. Anyiam, Kingsley O. Enalume, Ijuptil B. Gabriel, Chinonyelum L. Oforka, Hamze G. Dahir, Salisu T. Fausat, Getrude P. Massawe, Debra U. Okeh, Benjamin Karikari, Samuel T. Aderonke, Olushola M. Awoyemi, Idowu A. Aneyo, Funmilayo V. Doherty.

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
