## [Decision Letter · Decision Letter 0]

5 Jan 2022

PONE-D-21-36946COVID-19 vaccines uptake: Public knowledge, awareness, perception and acceptance among adult Africa nsPLOS ONE

Dear Dr. Ahiakpa,

Thank you for submitting your manuscript to PLOS ONE. After careful consideration, we feel that it has merit but does not fully meet PLOS ONE’s publication criteria as it currently stands. Therefore, we invite you to submit a revised version of the manuscript that addresses the points raised during the review process.

We look forward to receiving your revised manuscript.

Kind regards,

Sanjay Kumar Singh Patel, Ph.D.

Academic Editor

PLOS ONE

Journal Requirements:

2. During our internal checks, the in-house editorial staff noted that you conducted research or obtained samples in another country. Please check the relevant national regulations and laws applying to foreign researchers and state whether you obtained the required permits and approvals. Please address this in your ethics statement in both the manuscript and submission information. In addition, please ensure that you have suitably acknowledged the contributions of any local collaborators involved in this work in your authorship list and/or Acknowledgements. Authorship criteria is based on the International Committee of Medical Journal Editors (ICMJE) Uniform Requirements for Manuscripts Submitted to Biomedical Journals - for further information please see here: https://journals.plos.org/plosone/s/authorship.

 [We had no funding for this study]. 

4. We note you have included a table to which you do not refer in the text of your manuscript. Please ensure that you refer to Table 4 in your text; if accepted, production will need this reference to link the reader to the Table.

Reviewers' comments:

Reviewer's Responses to Questions

**Comments to the Author**

1. Is the manuscript technically sound, and do the data support the conclusions?

Reviewer #1: Yes

Reviewer #2: Yes

2. Has the statistical analysis been performed appropriately and rigorously? 

Reviewer #1: Yes

Reviewer #2: Yes

3. Have the authors made all data underlying the findings in their manuscript fully available?

Reviewer #1: Yes

Reviewer #2: Yes

4. Is the manuscript presented in an intelligible fashion and written in standard English?

Reviewer #1: Yes

Reviewer #2: Yes

5. Review Comments to the Author

Reviewer #1: In this paper entitled "COVID-19 vaccines uptake: Public knowledge, awareness, perception and acceptance among adult Africa ns", the authors investigated adult African inhabitant's views toward vaccination and their participation possibility in governments efforts to get citizens vaccinated. Adult Africans' responses were taken in the manuscript using a cross-sectional online survey, and the factors associated with willingness to accept the COVID-19 vaccine and participation were evaluated. The results indicated that more than two-thirds of African adults would not receive the COVID-19 vaccine as they have skepticism about it. The manuscript is easy to understand and technically correct. Furthermore, the manuscript is statistically sound and has potential. Therefore, it may be considered for publication after minor corrections.

Minor comments:

1) The English may be improved for the manuscript.

2) In the title, define adult Africa ns?. The authors should cross-check all abbreviations in the manuscript. Initially, define in the full name followed by abbreviations.

3) Introduction section may be minor polished with information such as - i) COVID-19 details, symptoms and prevention strategies including health status and diet i.e. doi: 10.1007/s12088-020-00908-0; doi: 10.1007/s12088-020-00893-4; ii) about COVID-19 variants and their future challenges i.e. doi: 10.1007/s15010-021-01734-2.

4) Design of the study and participants: How the author determines the number of participants?. Which reference study or formula is used to calculate the participant number for the study?.

5) Introduction, the importance of this study may be more specifically highlighted.

7) The author may provide a paragraph regarding challenges or prospects of study in the discussion and provide a limitation of the study.

Reviewer #2: 1. Various symptoms and prevention strategy of Covid-19 should be provided.

2. The variant of Covid-19 are measure concern for the treatment of infected patients. so, few more information may be provide.

3. Please provide include the data how the various vaccine are effective for the treatment of Covid-19 and their variant in the discussion section.

4. please illustrate or highlighted the summary for the significant of present study (Add 1or 2 figures).

5. Please combine 1-3 figures in one figure as they are very small.

---

## [Author Response · Author response to Decision Letter 0]

31 Jan 2022

Dear Editor,

We are grateful to you, and the rreviewers for giving us the opportunity to improve the scientific quality of our manuscript. We have revised the manuscript and prepared point-by-point responses to the reviewers’ comments. We look forward to your decision soon.

Thank you.

Journal Requirements:

#Authors’ response: We revised the sections are recommended.

2. During our internal checks, the in-house editorial staff noted that you conducted research or obtained samples in another country. Please check the relevant national regulations and laws applying to foreign researchers and state whether you obtained the required permits and approvals. Please address this in your ethics statement in both the manuscript and submission information. In addition, please ensure that you have suitably acknowledged the contributions of any local collaborators involved in this work in your authorship list and/or Acknowledgements. Authorship criteria is based on the International Committee of Medical Journal Editors (ICMJE) Uniform Requirements for Manuscripts Submitted to Biomedical Journals - for further information please see here: https://journals.plos.org/plosone/s/authorship.

#Authors’ response: The study was conducted online using the chain-referral approach where participants voluntarily participated in the survey. Consortium members/co-authors in the respective countries didn’t have any influence in the participation of the study. Thus, the voluntary nature and non-geographical specificity of the study preclude from obtaining ethical clearance from each country. Thus, a recognised ethical clearance from the study’s origin, Somalia should be adequate.

 [We had no funding for this study]. 

#Authors’ response: We didn’t receive any funding from any agency.

#Authors’ response: There is was no funding for this study

#Authors’ response: This is not applicable

#Authors’ response: The authors received no specific funding for this work

4. We note you have included a table to which you do not refer in the text of your manuscript. Please ensure that you refer to Table 4 in your text; if accepted, production will need this reference to link the reader to the Table.

#Authors’ response: We have cited the table in the revised manuscript. See L268-L300 in the revised manuscript.

#Authors’ response: We have reviewed the reference list and updated it where necessary

Response to Review Comments 

Dear Editor,

We are submitting the revised version of our manuscript with point-by-point responses to the reviewers’ comments. We are grateful to you, and the reviewers for the opportunity to improve the scientific quality/rigour of our manuscript. Below are point-by-point responses to issues raised by the reviewers and yourself. Corrections were directly effected in the manuscript in track changes. 

Thank you.

Reviewers' comments:

Reviewer's Responses to Questions

Comments to the Author

1. Is the manuscript technically sound, and do the data support the conclusions?

Reviewer #1: Yes

Reviewer #2: Yes

#Authors’ response: Thank you for your assessment and comments

2. Has the statistical analysis been performed appropriately and rigorously? 

Reviewer #1: Yes

Reviewer #2: Yes

 #Authors’ response: Thank you for your comments

3. Have the authors made all data underlying the findings in their manuscript fully available?

Reviewer #1: Yes

Reviewer #2: Yes

 #Authors’ response: Thank you for the comments

4. Is the manuscript presented in an intelligible fashion and written in standard English?

Reviewer #1: Yes

Reviewer #2: Yes

#Authors’ response: We appreciate the comments

 5. Review Comments to the Author

Reviewer #1: In this paper entitled "COVID-19 vaccines uptake: Public knowledge, awareness, perception and acceptance among adult Africans", the authors investigated adult African inhabitant's views toward vaccination and their participation possibility in governments efforts to get citizens vaccinated. Adult Africans' responses were taken in the manuscript using a cross-sectional online survey, and the factors associated with willingness to accept the COVID-19 vaccine and participation were evaluated. The results indicated that more than two-thirds of African adults would not receive the COVID-19 vaccine as they have skepticisms about it. The manuscript is easy to understand and technically correct. Furthermore, the manuscript is statistically sound and has potential. Therefore, it may be considered for publication after minor corrections.

#Authors’ response: We appreciate the comments and recommendation

Minor comments:

1) The English may be improved for the manuscript.

#Authors’ response: We have revised the manuscript where necessary.

2) In the title, define adult Africans?. The authors should cross-check all abbreviations in the manuscript. Initially, define in the full name followed by abbreviations.

#Authors’ response: We defined the category of participants in the study in the methodology section. See L139 in the revised manuscript.

3) Introduction section may be minor polished with information such as - i) COVID-19 details, symptoms and prevention strategies including health status and diet i.e. doi: 10.1007/s12088-020-00908-0; doi: 10.1007/s12088-020-00893-4; ii) about COVID-19 variants and their future challenges i.e. doi: 10.1007/s15010-021-01734-2.

#Authors’ response: We defined the category of participants in the study in the methodology section. See L139 in the revised manuscript. We have equally updated the introduction with the specific recommendations. See L79-94; L100-L107; and L126-144 in the revised manuscript.

4) Design of the study and participants: How the author determines the number of participants? Which reference study or formula is used to calculate the participant number for the study?

#Authors’ response: The study was conducted using random selection process with a cross-sectional sampling (snowballing) or chain-referral approach where participants voluntarily participated in the survey. The conventional Cochran formula [29] was used to determine the starting sample size;

no = \\frac{Z2pq}{e2},

where e = the desired precision level (margin of error), where p is the fraction of population, q is 1-p, and Z is the Z-value found in a Z table. A total of 365 participants completed the closed-ended questionnaire for our study. At a 95% level of confidence, this corresponds to a 2 % margin of error [29]. See L210-227

5) Introduction, the importance of this study may be more specifically highlighted.

#Authors’ response: We stated the relevance of the study specifically in the introduction. See lines 111-129.

7) The author may provide a paragraph regarding challenges or prospects of study in the discussion and provide a limitation of the study.

#Authors’ response: We have rephrased this in the discussion section as recommended. See L495-501

Reviewer #2: 

1. Various symptoms and prevention strategy of Covid-19 should be provided.

#Authors’ response: We have incorporated this in the revised manuscript

2. The variant of Covid-19 are measure concern for the treatment of infected patients. so, few more information may be provide.

#Authors’ response: We have included information on this in the revised manuscript

3. Please provide include the data how the various vaccine is effective for the treatment of Covid-19 and their variant in the discussion section.

#Authors’ response: We provided percentage efficacies of the various vaccines in the introduction. See lines 126-137.

4. please illustrate or highlighted the summary for the significant of present study (Add 1or 2 figures).

#Authors’ response: We have incorporated this in the revised manuscript. See L495-501.

5. Please combine 1-3 figures in one figure as they are very small.

#Authors’ response: We have merged these figures as recommended. Thank you.

---

## [Decision Letter · Decision Letter 1]

26 Apr 2022

COVID-19 vaccines uptake: Public knowledge, awareness, perception and acceptance among adult Africans

PONE-D-21-36946R1

Dear Dr. Ahiakpa,

We’re pleased to inform you that your manuscript has been judged scientifically suitable for publication and will be formally accepted for publication once it meets all outstanding technical requirements.

Kind regards,

Carla Pegoraro

Division Editor

PLOS ONE

Reviewers' comments:

Reviewer's Responses to Questions

**Comments to the Author**

1. If the authors have adequately addressed your comments raised in a previous round of review and you feel that this manuscript is now acceptable for publication, you may indicate that here to bypass the “Comments to the Author” section, enter your conflict of interest statement in the “Confidential to Editor” section, and submit your "Accept" recommendation.

Reviewer #1: All comments have been addressed

Reviewer #2: (No Response)

Reviewer #3: All comments have been addressed

2. Is the manuscript technically sound, and do the data support the conclusions?

Reviewer #1: Yes

Reviewer #2: Yes

Reviewer #3: Yes

3. Has the statistical analysis been performed appropriately and rigorously? 

Reviewer #1: Yes

Reviewer #2: Yes

Reviewer #3: Yes

4. Have the authors made all data underlying the findings in their manuscript fully available?

Reviewer #1: Yes

Reviewer #2: Yes

Reviewer #3: Yes

5. Is the manuscript presented in an intelligible fashion and written in standard English?

Reviewer #1: Yes

Reviewer #2: Yes

Reviewer #3: Yes

6. Review Comments to the Author

Reviewer #1: In this paper entitled "COVID-19 vaccines uptake: Public knowledge, awareness, perception and acceptance among adult Africans ", the authors have addressed all the comments and have no technical deficiency for rejection. The paper is eligible for acceptance in the journal.

Reviewer #2: The authors have revised the manuscript carefully. So, I think it can be accepted for publication.

Reviewer #3: (No Response)

7. PLOS authors have the option to publish the peer review history of their article (what does this mean?). If published, this will include your full peer review and any attached files.

Reviewer #1: **Yes: **Aditya Kumar Sharma

Reviewer #2: No

Reviewer #3: No

---

## [Editor Report · Acceptance letter]

5 May 2022

PONE-D-21-36946R1 

COVID-19 vaccines uptake: Public knowledge, awareness, perception and acceptance among adult Africans 

Dear Dr. Ahiakpa:

I'm pleased to inform you that your manuscript has been deemed suitable for publication in PLOS ONE. Congratulations! Your manuscript is now with our production department. 

Kind regards, 

on behalf of

Dr Carla Pegoraro 

Staff Editor

PLOS ONE